# Longitudinal study of calf morbidity and mortality on smallholder farms in southern Ethiopia

**Ephrem Tora** [ID]*, **Edget Abayneh, Wasihun Seyoum, Mesfin Shurbe**

Department of Animal Science and Health, College of Agricultural Sciences, Arba Minch University, Arba Minch, Ethiopia

* adech.tor@gmail.com

## Abstract

Calf morbidity and mortality are serious constraints in the success of dairy calf production. Thus, the current study was carried out with the objective to estimate the incidence of calf morbidity and mortality and associated risk factors in milk-shed districts of Gamo Zone, Southern Ethiopia. A prospective cohort and cross-sectional survey were employed from November 2019 to April 2020. A total of 196 calves were recruited by simple random sampling. Recruitment of calves was deployed by both the concurrent and prospective cohorts in calves aged below three months in study herds. The crude incidence of calf morbidity and mortality was 30.9% and 8.64%, respectively. The most frequently encountered disorder was calf diarrhea (10.17%), followed by pneumonia (6.5%). The other disorders were septicemia, omphalitis, arthritis, eye problem and miscellaneous cases. Multivariable Cox regression was revealed significant association for the calf vigor status, colostrum ingestion time, colostrum feeding status, dam parity, age at first calving, and related disorders were found risk factors of calf morbidity; likewise, calf vigor status at birth, time of colostrum ingestion and weaning were risk factors determining calf mortality. Calf morbidity and mortality rates recorded in this study were marginally higher than economically tolerable level, therefore, could affect the productivity of smallholder dairying by decreasing the obtainability of replacement heifers. Among significant explanatory factors investigated, colostrum ingestion time, method and amount were found important determinant factors of calf mortality and morbidity under the small-holder farming in the milk-shed districts of the Gamo zone. Therefore, rigorous calf husbandry practice is a need to manipulate the aforementioned calf determinants with subsequent application of tailor-made interventions.

## Introduction

Ethiopia has the largest livestock population, being first in Africa and tenth in the world [1, 2]. The country has diverse climatic condition, and livestock production takes place across all agro-ecological zones where the mixed small scale farming is predominant [3]. There is also intensive livestock production systems that accounted for market oriented production [4, 5].

**Data Availability Statement:** All the required datum are available and recorded in the format of excel we have attached as Supporting information files.

**Funding:** No grant is provided.

**Competing interests:** The authors declare that they have no conflict interests.

The significant role of livestock production in the national economy as well as in the socioeconomic development of millions of rural smallholder farmers has considerable prospective opportunity for income generation, employment and poverty alleviation [6, 7]. Thus, the sector contributes 15% of export earnings and currently supports and sustains livelihoods for 80% of all rural population [8].

Although the expansion of the dairy segment in Ethiopia has a considerable prospective opportunity for dairy producers' food and nutrition security [9], dairy production is severely affected by several constraints. Among the constraints, calf diseases and pre-weaning mortality are the major ones that determine successful rearing of replacement stock [7, 10]. Calf mortality incurs a great economic loss to dairy producers. This arises from deaths, treatment costs, decreased lifetime productivity and longevity. It also causes the loss of genetic material for herd improvement and decreases the number of dairy heifers available for herd replacement and expansion [11], leading to a shortage of dairy replacement heifers [12].

Over the years, a number of researchers have reported the occurrence of calf morbidity and mortality in Ethiopia. According to the observational studies, the apparent prevalence of calf mortality estimate falls within the range of 0.9% and 37.3%, whereas the calf morbidity estimate falls within the range of 22.8% and 66.7% [2, 10, 13–17]. This shows the current apparent prevalence range lies on a calf mortality rate of 20% that can reduce net profit by 38% [18, 19]. Also, calf mortality was ranked next to mastitis as the second main problem for dairy production in Ethiopia [20, 21]. Besides, the aforementioned studies have shown that different factors such as breed of the calf, age of calf, age at first colostrum ingestion and others contribute to the variations of apparent prevalence of calf mortality. However, to date there is scarcity of reports on the prevalence and the important host, management, and environmental risk factors influencing the occurrence of calf morbidity and mortality in the study area. Such information could be of significant importance in the development of feasible intervention measures aimed at reducing the burden of deaths and productivity losses. Therefore, the study was designed to estimate the incidence of calf morbidity and mortality and the associated risk factors in urban and peri-urban areas of milk-shed districts of the Gamo zone, Southern Ethiopia.

## Materials and methods

### Study area description

The study was conducted in urban and peri-urban smallholder dairy farms of Gamo zone milk-shed Districts. It includes a total of four administrative districts (Chencha, Kamba, Geresse, and Bonke districts). In these areas, hundreds of smallholder farmers produce milk from cross bred and indigenous cattle breeds. The largest production system in the areas is mixed crop-livestock farming and cattle are the most important livestock species reared in these areas. According to the Gamo zone livestock and fishery resource department (2020) [22], the estimated cattle population of the zone is 1,297,173. The altitude of Chencha district is between 1300 and 3250 meters above sea level. The astronomical location of Chencha district is between 37° 29' 57" East to 37° 39' 36" West longitude and between 6 °8'55" North and 6° 25'30" South latitude. Due to a high altitudinal range, the area is characterized by diverse agro-climatic distribution. The annual rainfall distribution ranges from 900 mm to 1200mm. The minimum temperature of the district ranges from 11°C—13°C, while the maximum temperature is in the range of 18°C—23°C. Geresse is one of the districts in the Zone and lies between 5°55′ N latitude and 37°15′ E longitude with an altitude ranging from 600–4200 masl. The mean annual average rainfall and the mean annual average temperature of the woreda are 1400 mm and 13.05°C, respectively. Bonke district lies between 5°55′ N latitude and 37°15′ E

longitude with an altitude ranging from 2200–4200 masl. Kemba district is located between 33˚53'E longitude and 6˚16'N latitude. Geographically, the area is located at an altitude of 1500 to 2500 meters above sea level. It receives mean annual rainfall of 700 to 2200 mm. the average annual temperature ranges from 22ºC to 28ºC and relative humidity is about 60% [1, 23].

## Study farms and population

The study was conducted in the urban and peri-urban smallholder dairy farms, which located in the milk sheds districts such as Bonke, Chencha, Geresse, and Kemba in the Gamo zone. The smallholder dairy farms located in the town are taken as urban dairy farms, which cover semi-intensive management and are engaged in market oriented dairy production. Dairy farms located on the outskirts of the respective towns are considered periurban dairy farms. Both Zebu and crossbred dairy calves of both sexes reared under smallholder dairy farms, and calves aged between births to 6 months were considered in the study. The inclusion criteria required that the calves were recruited into the study within birth to 3 months of age. However, still born calves, calves with deformities and wasting diseases, as well as calves that were not born in selected urbans and outskirts were excluded from this study. Following recruitment into the study, routine monitoring of study calves was done at 2 week intervals until 6 months old.

## Study design

The study was conducted comprising a cross sectional survey and prospective cohort study. The cross sectional survey was conducted using questionnaire and field observation for the assessment of associated risk factors on calf morbidity and mortality. The questionnaire survey was carried out by interview to smallholder dairy producers during the study period to obtain herd-level data relating to farm characteristics, calf and dam factors, management, health and environmental associated risk factors, and pre-parturient cow management practices. It used to support the prospective cohort results.

**Sampling technique and sample size determination.** Initially, using purposive sampling techniques, the study districts were selected based on the dairy potential. Accordingly, four districts from the Gamo zone namely Bonke, Chencha, Geresse, and Kemba were selected. Then, two areas (urban and peri-urban) from each district were selected purposively based on the availability of calf at birth and before three months of age, and presumed cooperativeness of the owners. Finally, the household heads having late term pregnant cows and/ or calves with age less than three months were selected and interviewed using a pretested semi-structured questionnaire. The total sample size for household interview was determined according to Arsham [24]. The sample size, N, could be expressed as integer less than or equal to $0.25/SE^2$; $N = \frac{0.25}{SE^2}$, Where SE is Standard Error and is 0.04 at 95% confidence interval; N is a number of samples. Approximately, 40 households were targeted from each district, and all calves fulfilling the inclusion criteria during the sample follow up period were randomly included in the study. Therefore, a total of 160 household heads/ calf attendants were employed. A prospective observational study was conducted between November 2019 to April 2020 to estimate the incidence of calf morbidity and mortality, and to assess the associated risk factors. The methods which were employed include farm visiting and enrolment of calves. Different works of literature have been proved that there is a difference in the incidence of morbidity and mortality between crossbred and local calves [16, 17] based on these previous studies on birth to six months of aged calves, the morbidity and mortality was 22.8% and 7.8%, in crossbred and local calves, respectively, was used to estimate calf morbidity and mortality. Based on the formula Habib [25] the optimal sample size for cohort study in breeds was used to determine a

birth to six months calves morbidity in the study area:-

$$n' = \frac{\left[ Z_{\frac{\alpha}{2}}\sqrt{(r+1)\bar{p}\bar{q}} - Z_\beta\sqrt{rp_1q_1 + p_2q_2} \right]^2}{rE^2}, \quad n_1 = \frac{n'}{4}\left[ 1 + \sqrt{\frac{2(r+1)}{n'r|E|}} \right]^2, \quad E = P_2 - P_1$$

$$n' = n_1, \; n_2 = rn_1 \rightarrow r = \frac{n_2}{n_1} \; and \; \bar{p} = \frac{p_1 + rp_2}{r+1}, \; q = 1 - p$$

Where $P_1$: proportion of morbidity among local breed (suppose unexposed i.e. local calves); $P_2$: proportion of morbidity among crossbred breed (suppose exposed i.e. crossbred calves); P: estimated average proportion of $p_1$ and $p_2$. At alpha (two-tailed) is 5% and the power is 80%. There were at least a total of 160 calves from 160 households. This sample size was proportionately allocated for the four districts. However, based on the above formulae a total of 196 calves that fulfilled the inclusion criteria were recruited randomly and enrolled. Thus, *n*1 was 54 and *n*2 was 142 with a total of 196, 54 local breeds, and 142 crossbred calves were sampled.

**Farm visiting and calf recruitment criteria.** A total of 196 newly born calves were sampled from small-holder dairy farms and followed for approximately 6 months. Calves less than 3 months of age at the initial visit with a known date of birth and disease history were recruited retrospectively and allowed to join the prospective cohort. Upon recruitment, the calves were given an identification number, date of birth, sex, breed, presence of delivery complications and colostrum delivery strategy to the calves were recorded. Furthermore, the dam's parity, breed, udder condition, health and milk yield were recorded and marked at the earliest farm visit, then regularly visited by the investigator as well as by the assigned enumerator until the calves reached 6 months of age. At each farm visit, calf management, housing, and sanitation situations were also observed. Health issues of the calves that occurred out of the visiting time were communicated by owners with investigators. Emergency visits were conducted whenever there were calls with to farms due to calf health problems. The sick calf was treated after the clinical investigation and the necessary sample was collected. When the calf loss occurred, the date and reason of loss were recorded at the subsequent farm visit. The individual-calf risk factor was documented by means of check-list forms provided by the investigator at the beginning of the study. When calves reached 6 months of age, they were withdrawn from the follow-up group. However, calves that diseased or died before 6 months of age were considered for the failure time. Thus, they were censored on the date of diseased, or died whichever occurred first. Calves that sold voluntarily before six months of age were considered as censor but not failure, so time event is not taken for survival analysis.

## Data management and statistical analysis

For the purpose of this study, total calf-days at risk were converted to calf months at risk, as the age of calf was defined up to 6 months. Moreover, to facilitate result comparisons with other findings, true rates calculated for mortality, morbidity and specific disease condition were derived into risk rates based on the formula (RR = 1-e-True Rate) [26]. STATA statistical software version 14 for Kaplan–Meier and Cox regression was used to run. First, Kaplan–Meier method was employed to estimate the hazard function of observed hazard differences for each of the explanatory variables with crude morbidity and mortality. Then, the probability of obtaining the observed hazard curves was valued by the Log rank test at $P \leq 0.05$. Furthermore, Cox proportional hazard model was used to evaluate the association between explanatory variables and survival up to 180 days of age. The association of individual risk factors with an outcome variable was screened by univariate Cox-regression. Next, those variables significantly associated with the outcome variable at ($P \leq 0.05$) in the univariate analysis were

**Table 1. Household and dairy performance characteristics.**

| Variables | Observation | Minimum | Maximum | Mean | Totals |
|---|---|---|---|---|---|
| Total family size | 160 | 1 | 13 | 6.15 | 984 |
| Herd size | 160 | 2 | 25 | 5.6 | 897 |
| Weaning age (crossbred)[a] | 159 | 3 | 7 | 4.65 | - |
| Weaning age (local breed)[a] | 160 | 4 | 37 | 9.98 | - |
| Milk yield amount (liters) | 116 | 8.6 | 28 | 18 | 6768 |

[a] = weaning age in months.

recruited for the further multivariable analysis using multiple Cox-regressions to assess the independent effect. Finally, explanatory variables that best explained the calf morbidity and mortality were selected by a backward stepwise selection. Starting with the least significant explanatory variable, each of the non-significant explanatory variables were removed until the estimated regression coefficients for all retained variables were significant (P ≤ 0.05). Thus, the main effects were included in the final model.

# Results

## Household characteristics of the respondents

The household and dairy performance characteristics of smallholder respondents are presented in Table 1. The household characteristics of farmers and their distribution in the study location are also shown in Fig 1. Among the total smallholder dairy producers interviewed in Gamo zone milk-shed districts, 65% were urban and 35% were peri-urban smallholder dairy farms, and 79.38% of households were male headed and 21.62% were female headed (Fig 1).

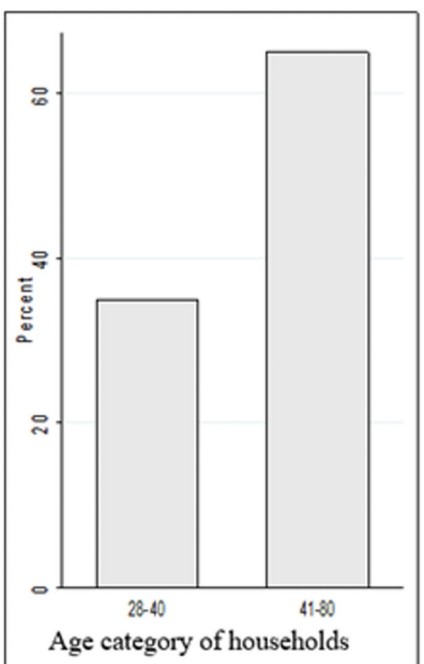
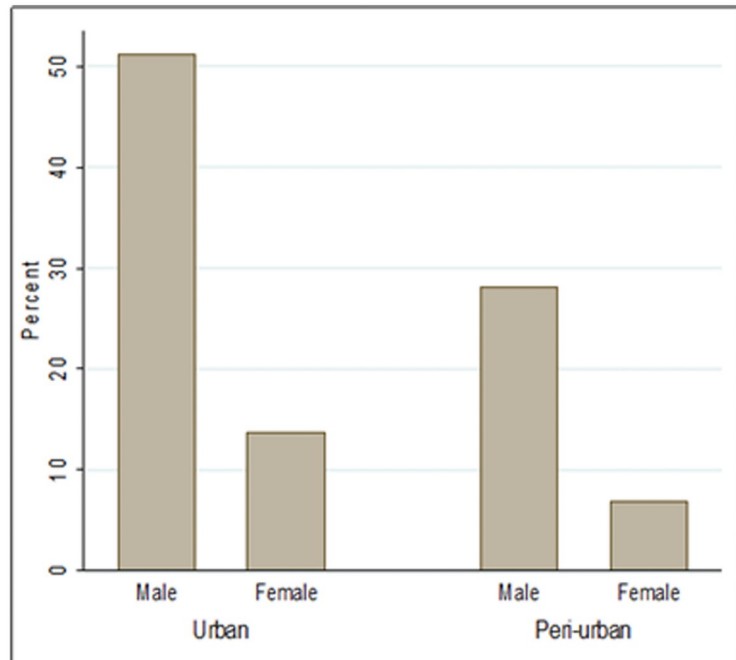

**Fig 1. The household characteristics of farmers and their distribution in the study location.**

**Table 2. Association of herd level household characteristics, farm management practices with calf health and mortality problems in farms within 1 year (N = 52).**

| Herd level Variables | Parameters | Frequency | Percent | $\chi^2$ | P-value |
|---|---|---|---|---|---|
| Education status | Illiterate | 32 | 20.00 | 2.90 | 0.234 |
|  | Literate | 128 | 80.00 |  |  |
| Dairy farm location | Urban | 104 | 65.00 | 4.76 | 0.093 |
|  | Peri-urban | 56 | 35.00 |  |  |
| Calf caretaker | Owner | 131 | 81.87 | 13.0 | 0.001 |
|  | Hired | 29 | 18.13 |  |  |
| Colostrum feeding awareness | Yes | 150 | 93.75 | 1.43 | 0.493 |
|  | No | 10 | 6.25 |  |  |
| Calf caretaker experiences | ≤ 5year | 35 | 21.88 | 34.7 | 0.000 |
|  | > 5year | 125 | 78.12 |  |  |
| Dairying serve as income | Primary | 55 | 34.38 | 4.40 | 0.110 |
|  | Secondary | 105 | 65.62 |  |  |
| Colostrum feeding quantity | Partial feeding | 65 | 40.63 | 16.9 | 0.000 |
|  | Complete feeding | 93 | 59.37 |  |  |
| Time of colostrum feeding | Before 6 hours | 107 | 66.88 | 16.6 | 0.000 |
|  | Between 6–24 hours | 53 | 31.12 |  |  |
| Colostrum feeding method | Suckling | 107 | 66.88 | 20.2 | 0.003 |
|  | Bucket feeding | 53 | 33.12 |  |  |
| Calving facility | Calving pen | 56 | 35.00 | 24.7 | 0.000 |
|  | In same burn | 104 | 65.00 |  |  |
| Frequency of calf pen cleaning | Daily | 142 | 88.75 | 5.44 | 0.245 |
|  | Twice a day | 18 | 11.25 |  |  |
| Mode of replace feeding | Free grazing | 12 | 7.50 | 52.3 | 0.000 |
|  | Stall feeding | 118 | 73.75 |  |  |
|  | Partial feeding | 30 | 18.75 |  |  |

AI = Artificial Insemination; N = Number of calf health related problems; χ2 = Chi-square.

With regards to respondents age, most (65%) of the respondents were in the age group of 41–80 years (Fig 1). The average herd size of the study farms in the urban and peri-urban dairy system was 5.6 cattle heads. In the present study, it was reported that dairy production was a primary business for 34.38% of farm owners (Table 2).

## Farm management characteristics and calf health related problems

The details on the smallholder respondents' overall feedback on their household and dairy farm management practices in association with calf health related problems within one year in the districts are presented in the Table 2. In this herd-level, questionnaire and farm observation, calf morbidity and mortality, birth-related disorders and mastitis were the major problems of smallholder dairy farms.

About 82% of calf husbandry practices were taken by owner calf caretakers. Majority of (82%) calf husbandry practices were carried out by owner. In the study districts, most of the calf caretakers (78.2%) had experience of greater than five years. Hence, there was a significant association (P < 0.05) between the calf health problems and respondents work experiences. Although smallholder farmers allowed calves to suckle and bucket fed colostrum, the duration from the birth to the time of colostrum feeding was not consistent. Of the newborn calves, 66.8% were allowed to suckle and/or bucket feed colostrum within 1 to 6 hours after birth.

**Table 3. Follow-up life table of 196 calves born in Gamo zone milk-shed districts.**

| Interval start time | Number entering this interval | Number exposed to risk | Deaths | Withdrawal | Proportion surviving | Proportion failure | Hazard rate |
|---|---|---|---|---|---|---|---|
| 0–30 | 196 | 196 | 4 | 8 | 0.9796 | 0.0208 | 0.0007 |
| 30–60 | 184 | 184 | 3 | 2 | 0.9636 | 0.0369 | 0.0006 |
| 60–90 | 179 | 179 | 1 | 0 | 0.9582 | 0.0423 | 0.0002 |
| 90–120 | 178 | 178 | 5 | 0 | 0.9313 | 0.0692 | 0.0009 |
| 120–150 | 173 | 173 | 2 | 0 | 0.9206 | 0.0799 | 0.0004 |
| 150–180 | 171 | 171 | 1 | 0 | 0.9152 | 0.0853 | 0.0002 |
| 180 - | 170 | 170 | 0 | 170 | 0.9152 | - | - |

## Prospective cohort study

A total of 196 calves were enrolled from smallholder dairy farms in urban and peri-urban areas of Gamo zone milk-shed districts. During the start of the study, calf with birth and history data were recruited concurrently were 45% of the total, and the remaining calves were resulted from births within the study time frame. Female and male calves attributed over the observation period were 108 (55%) and 88 (45%) respectively. These numbers of calves contributed a total of 28,043 and 32,012 calf-days, and 77.9 and 88.9 calf-years at risk for morbidity and mortality respectively (Tables 4 and 5).

The temporal pattern of the study cohort is shown (Table 3). A total of 58 (24.5%) calves, 24 female and 34 male, were exited the study period due to diseases, deaths and sales. The censor rate due to deaths and sales was 13.3%. The majority of calves exiting were male calves that were sold because small holders did not want male calves since they incur extra-feeding cost and are less valuable than female calves.

## Estimates of morbidity and mortality

**Incidence of morbidity.** The crude morbidity and cumulative incidences of various disease conditions/ syndromes recorded in the study is presented in Table 4. Among syndromes diagnosed during the follow-up period, calf diarrhea was the greatest factor associated with morbidity (10.17%). The risk rate for pneumonia, arthritis, navel illness and septicemic condition were all between 0.64% and 6.55%.

The log rank test for differences in Kaplan-Meier curves of the calf health disorders was statistically significant (p-value = 0.001). The median ages of calves for the incidence of morbidity

**Table 4. The incidence of different diseases and disease conditions/syndromes in calves.**

| Diseases condition/ syndromes | Number of cases | Calf days at risk | Per calf month | Calf six months at risk | Incidence | |
|---|---|---|---|---|---|---|
| | | | | | True rate (6 calf month at risk) | Risk rate (%)[a] |
| Diarrhea | 16 | 26839 | 894.6 | 149 | 0.107 | 10.17 |
| Pneumonia | 10 | 27455 | 915.2 | 152 | 0.065 | 6.55 |
| Navel illness | 4 | 28004 | 933.5 | 155 | 0.025 | 2.57 |
| Arthritis | 3 | 27699 | 923.3 | 154 | 0.019 | 1.91 |
| Septicaemic condition | 8 | 27589 | 919.6 | 153 | 0.052 | 5.22 |
| Eye problem | 1 | 27954 | 931.8 | 155 | 0.006 | 0.64 |
| Miscellaneous cases | 6 | 27653 | 921.7 | 153 | 0.039 | 3.91 |
| Crude morbidity | 48 | 28043 | 934.7 | 156 | 0.365 | 30.9 |

[a] = Risk rates estimated from true rate using formula, Risk rate = $1-e^{-true\ rate}$ [26].

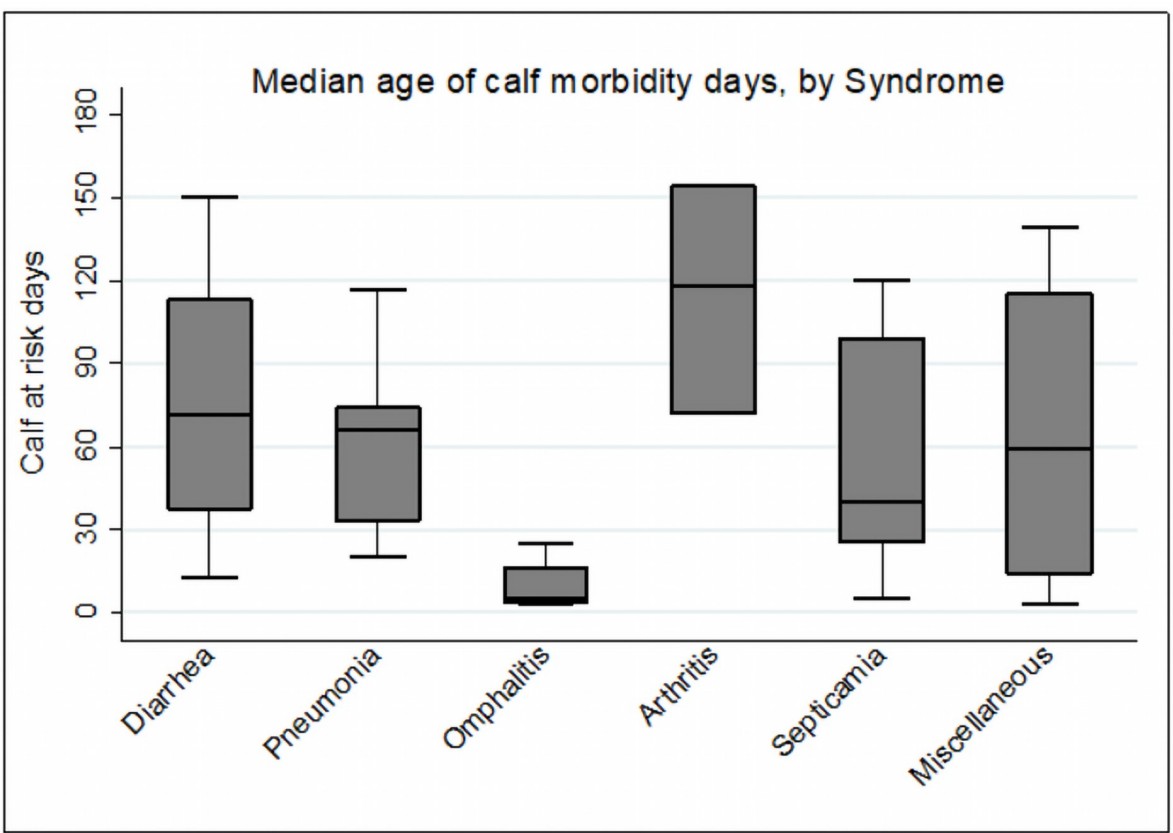

**Fig 2. The description of the median age of morbidity by syndromes.**

were 64.5 days. The median survival time for disease conditions: diarrhea, pneumonia, septicemia, and navel illness were 71.5, 66, 40 and 5.5 days, respectively (Fig 2). These finding indicated that disease commonly occurs in calves less than 3 months of age, with 10.7% morbidity occurring in the first week of life. Cumulatively, 27% morbidity occurred in the first month of life and 72.9% occurred in the first 3 months of life.

**Incidence of mortality.** The incidence of calf mortality associated with different diseases is shown in Table 5. The crude calf mortality in the present study was 8.64%. In this prospective study, median ages of calves for the incidence of mortality were 75 days. The principal disease syndrome associated with calf mortality was septicaemic condition, directly accounting

**Table 5. The mortality of calf associated with different diseases.**

| Cause of death | Number of cases | Calf days at risk | Per calf month | Calf six months at risk | Incidence | |
|---|---|---|---|---|---|---|
| | | | | | True rate (6 calf months at risk) | Risk rate (%) |
| Diarrhea | 4 | 29416 | 980.53 | 163 | 0.02447 | 2.42 |
| Pneumonia | 1 | 30348 | 1011.6 | 169 | 0.00593 | 0.59 |
| Navel illness | 2 | 30972 | 1032.4 | 172 | 0.01162 | 1.16 |
| Septicemic condition | 6 | 31254 | 1041.8 | 174 | 0.03456 | 3.40 |
| Miscellaneous cases | 3 | 31290 | 1043.1 | 174 | 0.01742 | 1.73 |
| Crude mortality | 16 | 32012 | 1067.1 | 178 | 0.08996 | 8.64 |

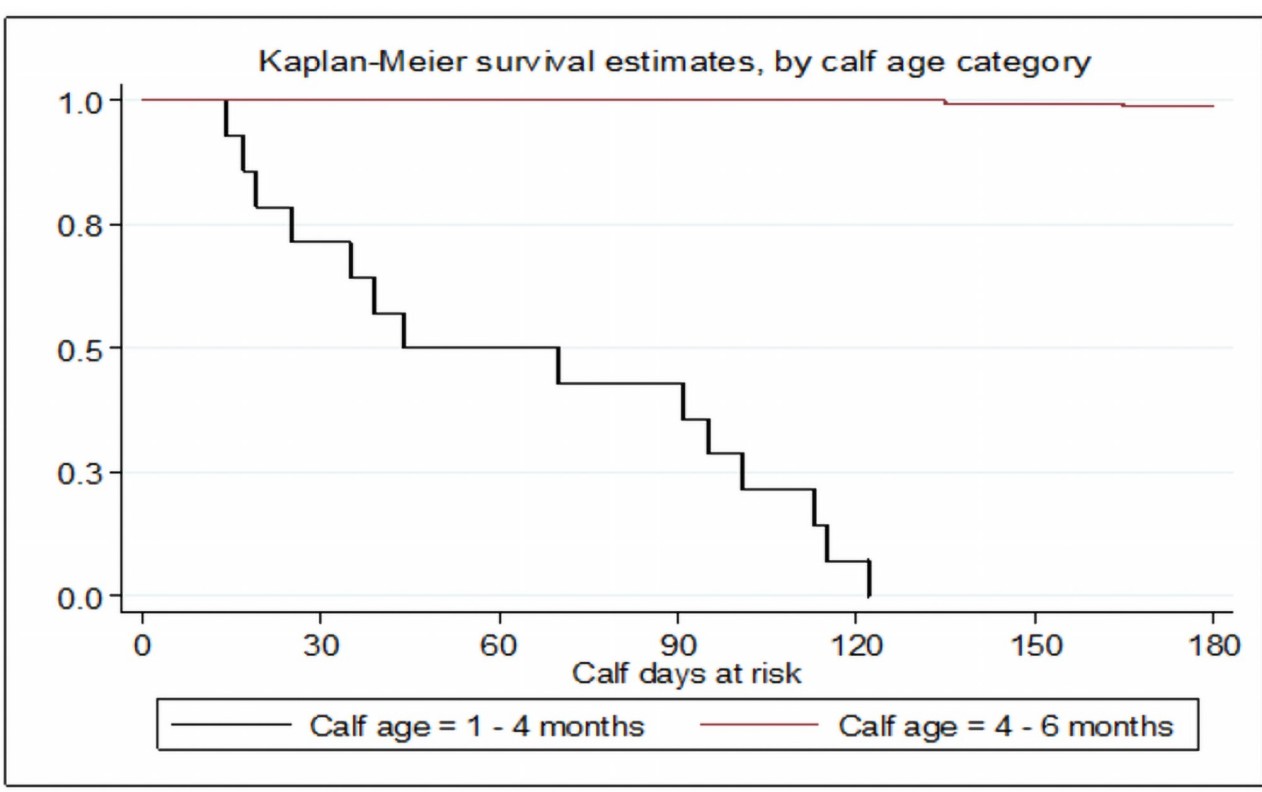

**Fig 3. Description of the median survival age of mortality by age category in months.**

for the 6 cases out of the 16 total deaths. The median survival time for disease conditions: septicaemic condition, diarrhea, pneumonia, and miscellaneous cases were 65, 118, 44 and 70 days, respectively. The incidence risk of calf mortality for disease disorders were 2.4, 3.4, 0.5, 1.7, and 1% diarrhea, septicemia, pneumonia, miscellaneous cases and navel illness, respectively (Table 5).

In this study, the median survival time for crude mortality was 75 days. This falls into the age category of 1–4 months shown in Fig 3. As a result, calf age related health events can be grouped into younger calves (<4 months), which is a critical age for calf health. In calves older than 4 months of age, the health events were less severe and declined gradually. Relatively higher risk of mortality in young calves age up to 4 months observed in this study was suggested the need of more careful management for very young calves as compared to older ones.

### Association of risk factors with incidence of morbidity and mortality

**Determinants of calf morbidity.** Univariate Cox regression analysis for the breed of calf was found significantly associated with calf morbidity (P ≤ 0.05) (Table 6). Essentially crossbred calves were at greater risk of morbidity than that of local breed calves. The temporal distribution of calf breeds morbidity rate across six months of study period presented in Fig 4, demonstrates the above findings.

Further, multivariable analysis using Cox regression by fitting into the model, the potential risk factors with significant association (P ≤ 0.05) were colostrum feeding status, vigor status at birth, time of the colostrum ingestion, dam parity, age at first calving and birth related disorders were found significantly associated with crude morbidity.

**Table 6. Potential determinants significantly associated with incidence of crude morbidity based on univariate analysis using Cox regression.**

| Variables | Categories | HR* | 95% CI for HR | P value |
|---|---|---|---|---|
| Calf breed | Cross vs. Local | 3.95 | 1.564–9.978 | 0.004 |
| Herd size | > 4 cattle vs. ≤ 4 cattle | 2.53 | 1.355–4.708 | 0.004 |
| Dairy as source of income | Secondary vs. Primary | 2.40 | 1.290–4.114 | 0.005 |
| Birth condition | Assisted vs. Easy | 2.25 | 1.118–4.511 | 0.023 |
| Assistance given | Owner vs. Veterinarian | 1.37 | 1.060–1.777 | 0.016 |
| Colostrum feeding adequacy | Partial vs. Complete | 3.88 | 1.981–7.625 | 0.000 |
| Colostrum fed status | No vs. Yes | 3.49 | 1.497–8.225 | 0.004 |
| Calf vigor status | Poor vs. Good | 4.39 | 2.229–8.651 | 0.000 |
| Time of colostrum ingestion | > 6hours vs. ≤ 6hours | 2.35 | 1.334–4.145 | 0.003 |
| Colostrum feeding method | Suckling vs. Bucket | 2.86 | 1.802–4.542 | 0.000 |
| Mothering instinct | Poor vs. Good | 2.74 | 1.083–6.942 | 0.033 |
| Dam breed | Cross vs. Local | 3.12 | 1.553–6.267 | 0.001 |
| Dam parity | Primiparous vs. Multiparous | 1.51 | 1.009–2.222 | 0.045 |
| Age at first calving | Continuous variable | 0.49 | 0.327–0.734 | 0.001 |
| Birth related disorders | Yes vs. No | 3.32 | 1.690–6.512 | 0.000 |
| Mastitis condition | Yes vs. No | 3.34 | 1.088–13.78 | 0.050 |
| Elevation (Altitude m.a.s.l) | >2000 vs. <2000 | 2.15 | 1.189–3.889 | 0.011 |

* Hazard ratio (which has similar meaning to relative risk).

In the present work, multivariable analysis of the risk factors found statistically associated with crude morbidity was age at first colostrum feeding and other risk factors (Table 7). The finding that delayed colostrum intake (> 6 hours of age) is associated with increased risk of morbidity. This indicated that the first six hours are the period in which ultimate absorption of colostral immunoglobulin takes place. Thus, higher risk of morbidity related to delayed intake of first colostrum meal could be associated with failure of passive transfer of colostral immunity.

In regards to the dam parity, the hazard of calf morbidity from calves born by primiparous cows had significantly (P ≤ 0.05) associated when compared to calves born from multiparous cows in the study. The association between calf vigor status and crude morbidity was evidenced in the hazard function curve (Fig 5).

**Determinants of crude mortality.** Table 8 displays the significant (P ≤ 0.05) factors associated with morbidity in the univariate Cox regression.

Among the variables significantly associated with crude calf mortality in the univariate analysis, only 3 of them were fitted into a model to run multivariable Cox regression. These were time of colostrum ingestion, weaning age, and calf vigor status at birth that contributed significantly to the model at 5% significant level (Table 9).

Keeping the effect of other factors constant, the hazard of mortality was 3.64 times higher for calves, which ingested their first colostrum later than 6 hours after birth than those ingested within 6 hours after birth. Hence, calves that had first colostrum after six hours of age had significantly increased mortality (P ≤ 0.05). The difference in hazard of mortality between calves grouped by most significant factor, age at first colostrum feeding, was evident in the hazard function curve (Fig 6).

This finding demonstrates a higher hazard of the mortality in weaned calves less than 4 months of age, when compared to calves greater than 4 months of age (P < 0.05). In this study, it was common practice for calves to be not fed dry feed prior to weaning age. Therefore,

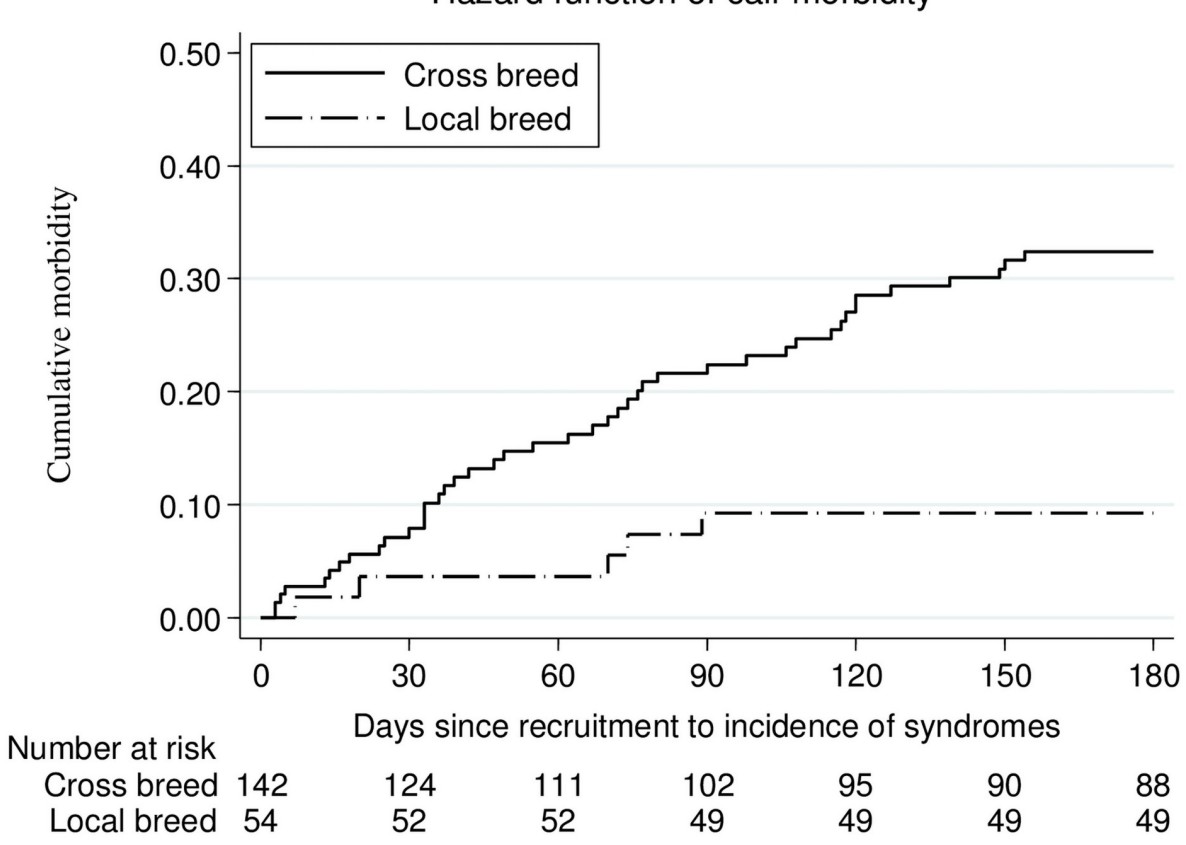

**Fig 4. Temporal distribution of breed morbidity factors incidences across 6 study months.**

calves failed to adjust to dry feed quickly and capably that leads to malnutrition and suscepti-bility to diseases in semi-intensive farms when weaned. However, calves in mixed production usually start fresh grass before weaning and were more likely to remain healthy.

## Discussion

It has been thought that, efficient dairy production and limited losses are important for farm-ers to achieve maximum productivity in dairy farming [27]. In this regard, calf morbidity and mortality is the most frustrating part involving in dairy industry and as such a terrible waste

**Table 7. Potential risk variables significantly associated with incidence of crude morbidity based on multivariable analysis using Cox regression.**

| Variables | Categories | HR* | 95% CI for HR | P value |
|---|---|---|---|---|
| Colostrum fed status | No vs. Yes | 7.32 | 2.232–18.41 | 0.001 |
| Calf vigor status | Poor vs. Good | 4.39 | 2.229–8.651 | 0.000 |
| Time of colostrum ingestion | > 6hours vs. ≤ 6hours | 2.36 | 1.099–2.946 | 0.018 |
| Dam parity | Primiparous vs. Multiparous | 1.87 | 1.162–3.015 | 0.010 |
| Age at first calving | Continuous variable | 0.38 | 0.215–0.684 | 0.001 |
| Birth related disorders | No vs. Yes | 2.31 | 1.916–6.842 | 0.045 |

* Hazard ratio (which has similar meaning to relative risk).

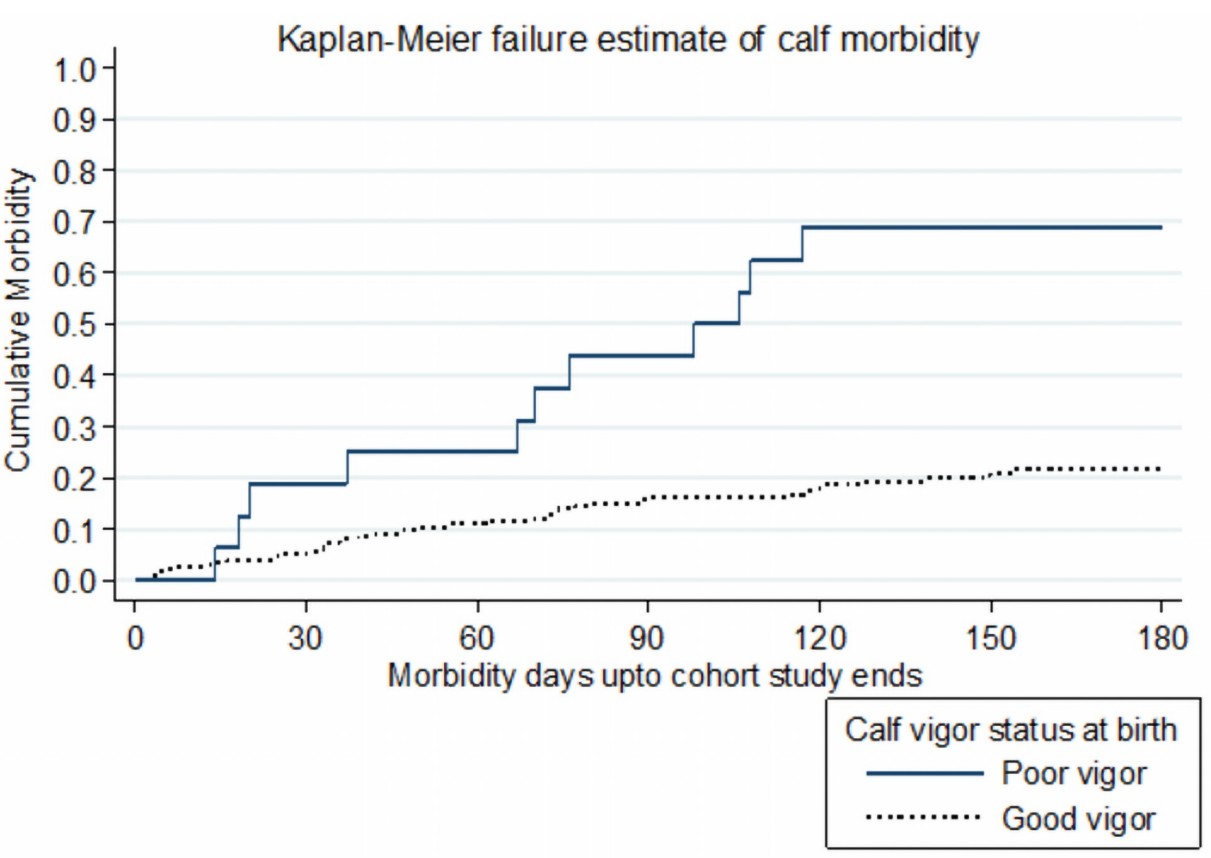

**Fig 5. Calf morbidity compared by calf vigor status at birth factor.**

and a killer of profit in which the producer has to wait another year before one can make up for the loss [28]. A greater appreciative of these causes of calf deaths and sickness patterns helps in identifying the major diseases and management related problems and dimensions for improvement, guiding research efforts, activities and decisions of the extension personnel, veterinarians, and policy makers in the management of dairy health in general and calf health in particular.

**Table 8. Potential risk variables significantly associated with the incidence of crude calf mortality based on univariate analysis using Cox regression.**

| Variables | Categories | HR* | 95% CI for HR | P value |
|---|---|---|---|---|
| Colostrum feeding adequacy | Partial vs. Complete | 4.54 | 1.293–15.92 | 0.018 |
| Colostrum fed status | No vs. Yes | 4.65 | 1.232–16.35 | 0.017 |
| Calf vigor status | Poor vs. Good | 7.91 | 2.869–21.82 | 0.000 |
| Time of colostrum ingestion | > 6hours vs. ≤ 6hours | 3.96 | 1.440–10.91 | 0.008 |
| Colostrum feeding method | Suckling vs. Bucket | 5.56 | 1.930–16.01 | 0.001 |
| Weaning age | > 4 months vs. ≤ 4 months | 4.42 | 2.582–15.71 | 0.000 |
| Mothering instinct | Poor vs. Good | 7.44 | 2.385–23.22 | 0.001 |
| Dam breed | Cross vs. Local | 3.24 | 1.012–12.50 | 0.048 |
| Dam parity | Primiparous vs. Multiparous | 1.51 | 1.009–2.222 | 0.045 |

* Hazard ratio (which has similar meaning to relative risk).

**Table 9. Potential risk factors significantly associated with the incidence of crude mortality based on multivariable analysis using Cox regression.**

| Variables | Categories | HR* | 95% CI for HR | P value |
|---|---|---|---|---|
| Time of colostrum ingestion | > 6hours vs. ≤ 6hours | 3.64 | 1.605–8.260 | 0.002 |
| Calf vigor status | Poor vs. Good | 3.10 | 1.071–13.34 | 0.013 |
| Weaning age | > 4 months vs. ≤ 4 months | 2.42 | 1.234–5.28 | 0.000 |

Within the urban and peri-urban dairy facilities of the districts of Gamo zone, the crude incidence risk of calf morbidity of 30.9% and calf mortality of 8.64% were recorded. This is relatively lower than previous report in Ethiopia. Most of the calf morbidity and mortality reports in the Ethiopia were based on studies in research stations, commercial and state farms with medium to large herd sizes and usually holding high exotic blood level animals, apparently these are associated with increased risk of calf disease occurrence [10, 13, 17, 29–32]. In the current study, the number of calves per smallholder dairy farms was small; therefore, each calf can be easily monitored and measures to avoid calf health problems. This could be an explanation as to why mortality rate should be less than those large herd size farms aforementioned. Thus, it is much higher than the economically tolerable level of 3–5% calf death [19, 33–35]. However, due to small number of death events which lower the power of the test in survival analysis, the results for that outcome might need cautious considerations.

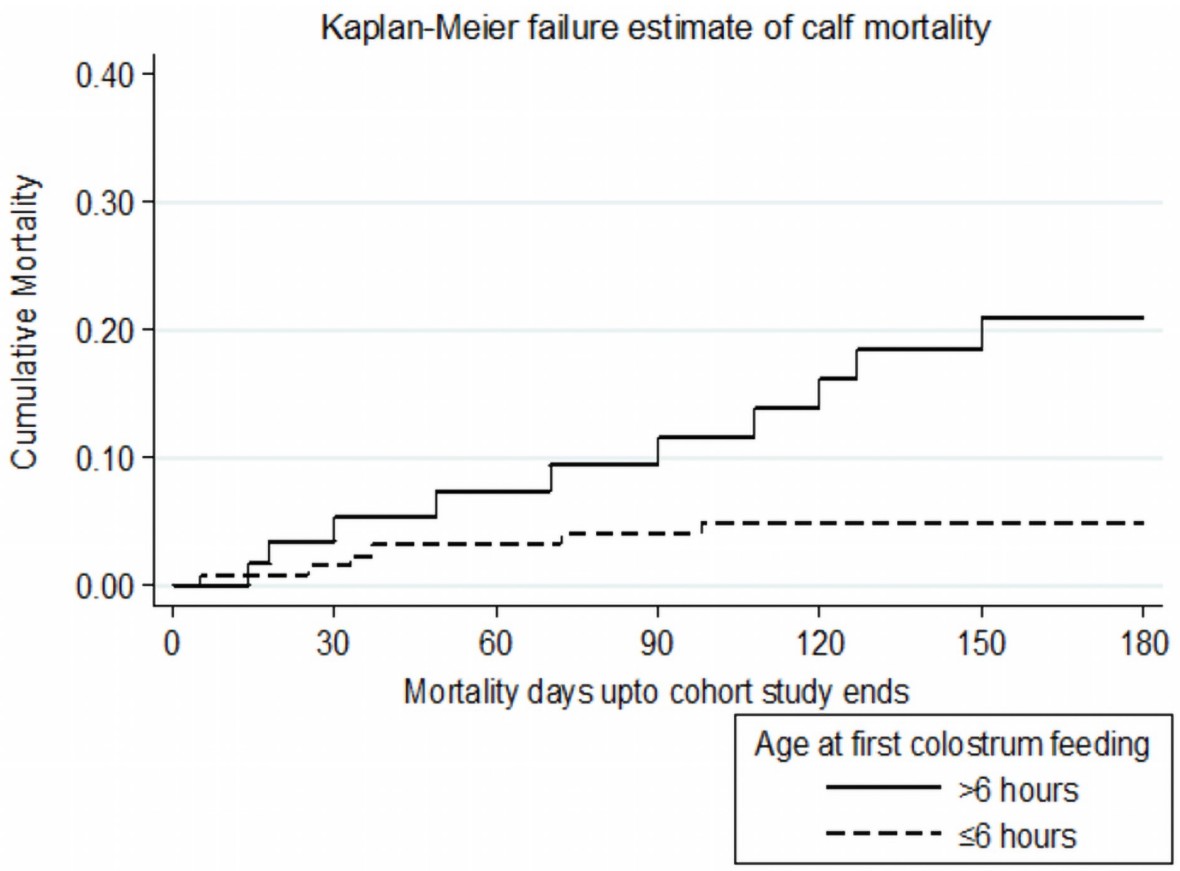

**Fig 6. Mortality compared by ages of first colostrum ingestion.**

The incidence of crude morbidity recorded over the six months follow-up period was 30.9%. This result is close to the findings of Bekele *et al.*, 2009 [15], where crude incidence rates of morbidity (29.3%). The result in the present study, however, is lower than the incidence rates of crude morbidity (62%) as reported by Wudu *et al.*, 2008 [10] and 66.7% morbidity as reported by Assefa and Ashenafi, 2016 [14] and 34.1% morbidity reported by Mohammed *et al.*, 2020 [4]. This difference might be due to better management system, better access to veterinary service in urban and its suburbs, small herd size, good awareness about importance of colostrum, most of the owners feed colostrum to their calves immediately. The latter difference might be due to education level.

Calf mortality is of the economic importance in dairy enterprise as this affects the number of calves that will survive for replacement herd [19]. A high calf mortality that ranges up to loss of about quarter of calves was observed in the urban and peri-urban dairy farms in Ethiopia [10, 36, 37]. The mortality rate reported in this study was 8.64% with the range (5.4–13.7). This finding is consistent with the previous calf mortality reports from Hawassa, which was 9.3% in the same range [15]. This result was closely comparable to earlier calf mortality studies in Ethiopia, such as the 6.5% and 6% calf mortality reported from Andasa ranch [13] and in central Ethiopia [17], respectively. However, this rate is very lower than calf mortality rates in dairy farming in Ethiopia. For instances, calf mortality of 20.9% in the Oromia region [36], and calf mortality of greater than 50% in Tropics [38] who had been insisted better calf management practices.

The risk of calf diarrhea was 10.43% in this study, which is comparable with the report of Bekele *et al.* [15] who reported 10%, Tsegaw *et al.* [36] who recorded 10.4%. In contrary, the present outcome was less than previous studies by Wudu *et al.* [10], 34%, Islam *et al.* [39], 34.8% and Assefa and Ashenafi [14], 42.9%. This finding has also been supported by other studies [10, 30], who documented the calf diarrhea as the commonest disease and utmost cause of neonatal morbidity during the first 3 months of life. Nonetheless, calf diarrhea as a prominent health problem in growing dairy calves, the lower incidence in this study suggests the significance of hygienic handling of feeding utensils and calf house observed during the study. Besides, based on interview and observation, most of the smallholder farmers were aware of the optimal time for colostrum feeding and this could greatly contribute to the lower incidence of calf diarrhea.

Among the explanatory variables that were categorized beneath farm attributes, most of that elucidate about personal attributes of people caring for calves were associated with health events. Similar reports have documented that such factors affect farm performance to greater or equal degree to other management variables [10, 30, 40, 41]. Likewise, the lower death losses due to mortality factors might be observed in farms where calves are managed by farm owner rather than managed by hired labor [42]. This finding has shown the importance of close relationship between the owner and the calves, and that this is likely to play a key role in improved animal welfare and health.

It has been reported that, early weaning is a risk factor for calf mortality in Ethiopia [4, 10, 13]. In the current study findings, calves have a higher risk of mortality when weaned below 4 months of age when compared to those weaned at 4 or greater than 4 months of age. However, the age supposed early and found to be risk factor for calves in present study was optimum age for weaning in the literatures in Ethiopia [5, 15]. Besides, calves stayed without being introduced to dry feed up to weaning age and no special starter feed was given in the study area. Based on this fact, weaned calves failed to adjust to dry feed quickly and efficiently that leads to malnutrition and susceptibility to diseases in semi-extensive farms. However, calves in mixed livestock and crop production start fresh grass before weaning. Generally, the age considered

early and found out to be risky for calves in the present study was within the range of the optimum age for weaning.

Keeping the effect of other factors constant, the hazard of mortality was 3.64 times higher for calves, which ingested their first colostrum later than 6 hours after birth than those ingested within 6 hours after birth. Hence, calves that had first colostrum after six hours of age had significantly increased mortality (P = 0.002). This finding was in agreement with the finding of Wudu *et al.* [10] who reported 2.24 times higher morbidity in calves ingested colostrum 6 hours later. As to morbidity association with delayed first colostrum feeding, these results are in agreement with other reports. Numerous previous reports have exposed that the first six hours of calf life is the period in which maximum absorption of colostral immunoglobulin takes place and higher risk of morbidity was related to failure of passive transfer of colostral immunity during this period [43, 44].

Though smallholder farmers allowed calves to suckle and bucket feed colostrum, the duration from the birth to time of colostrum feeding was not consistent. Of the newborn calves, 69.4% were offered suckle and bucket feeding of colostrum within 1 to 6 hours after birth and all farmers allowed colostrum feeding immediately after release of placenta. This result is similar to the reports of young stock mortality in different regions in Ethiopia [36]. Contrary to this result, other study, calves were fed partially or completely colostrum 1 to 2 hours without considering the release of placenta in India [45]. Urban and peri-urban smallholder farmers in the study districts, where most are literate were aware about disease resistance property of colostrum; hence majority of them allowed colostrum feeding of calves as soon as possible after birth.

## Conclusions

Dairy production is a primary business for 34.4% smallholders as resources of income generation and alleviation of poverty in milk-shed districts producing dairy calves. The calves morbidity and mortality rates were revealed higher in the present study area of urban and peri-urban dairy farms. This level was higher than economically tolerable for most smallholders. More significantly, smallholders have smaller herd sizes and the animals are a significant part of livelihood. Therefore, these higher rates of morbidity and some mortality decreased replacement herd numbers and ultimately hinder the success of smallholder dairy business. Calf diarrhea was the main calf health problem, whereas septicaemic conditions for deaths. The critical time for higher calf morbidity and mortality in this study was during the first 3 months of life. Moreover, this study revealed malpractices in calf management among smallholder owners, including malnutrition, restricted colostrum feeding, poor care and management, especially for calves in terms of milk allowance, supplemental feeding, and health management. However, most of the host and calf caretakers are responsive to suggested intervention. Therefore, making interventions against these risk factors could undoubtedly improve calf health status and minimize shortage of replacement heifers.

## Supporting information

**S1 Dataset.**
(XLSX)

**S2 Dataset.**
(XLSX)

## Acknowledgments

The authors would like to extend their deepest appreciation to their study participants for willingness to participate in the study and allowing us to conduct our study on their dairy farms. And, Dr. Teshale Sori from Addis Ababa University for all-embracing scientific guidance, valuable comments and discussion in all aspects of the study.

## Author Contributions

**Data curation:** Ephrem Tora.

**Formal analysis:** Ephrem Tora.

**Investigation:** Ephrem Tora, Edget Abayneh, Wasihun Seyoum, Mesfin Shurbe.

**Methodology:** Ephrem Tora, Edget Abayneh, Wasihun Seyoum, Mesfin Shurbe.

**Software:** Ephrem Tora, Mesfin Shurbe.

**Writing – original draft:** Ephrem Tora.

**Writing – review & editing:** Ephrem Tora.

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
