## [Decision Letter · Decision Letter 0]

25 May 2021

PONE-D-21-01541

LONGITUDINAL STUDY ON CALF MORBIDITY AND MORTALITY IN MILK-SHED DISTRICTS OF SOUTHERN ETHIOPIA

PLOS ONE

Dear Dr. Tora,

Thank you for submitting your manuscript to PLOS ONE. After careful consideration, we feel that it has merit but does not fully meet PLOS ONE’s publication criteria as it currently stands. Therefore, we invite you to submit a revised version of the manuscript that addresses the points raised during the review process.

Both expert reviewers have highlighted several concerns with your manuscript that preclude its acceptance as it stands. Having read your submission, I fully concur with the reviewers' comments and you must carefully address all comments when/if revising your manuscript. Furthermore, the English language of the manuscript requires extensive editing as it does not meet the minimum standards for publication. It is recommended that you enlist the help of an English-speaking colleague with experience in academic writing or use professional services.

We look forward to receiving your revised manuscript.

Kind regards,

Angel Abuelo, DVM, MRes, MSc, PhD, DABVP (Dairy), DECBHM

Academic Editor

PLOS ONE

Journal Requirements:

2. Thank you for submitting the above manuscript to PLOS ONE. During our internal evaluation of the manuscript, we found significant text overlap between your submission and the following previously published works.

- https://www.hindawi.com/journals/vmi/2020/6490710/ (background, paragraph 1, sentence 2) (background, paragraph 2, sentences 1-2) (background, paragraph 3, sentence 1) (background, paragraph 3, sentence 3-6)

- https://link.springer.com/article/10.1007/s11250-007-9104-3 ) (background, paragraph 7, sentence 8) (background, paragraph 8, sentence 1) (background, paragraph 8, sentences 4-5)

- https://cgspace.cgiar.org/bitstream/handle/10568/77368/thesis_yeshwas_2015.pdf?isAllowed=y&sequence=1 conclusion, paragraph 1, sentence 2)

- https://www.hindawi.com/journals/vmi/2020/3075429/ (conclusion, paragraph 1, sentences 6-7)

Please revise the manuscript to rephrase the duplicated text, cite your sources, and provide details as to how the current manuscript advances on previous work. Please note that further consideration is dependent on the submission of a manuscript that addresses these concerns about the overlap in text with published work.

5. We note that Figure 1 in your submission contains map images which may be copyrighted.

We require you to either (a) present written permission from the copyright holder to publish this figure specifically under the CC BY 4.0 license, or (b) remove the figure from your submission:

b. If you are unable to obtain permission from the original copyright holder to publish this figure under the CC BY 4.0 license or if the copyright holder’s requirements are incompatible with the CC BY 4.0 license, please either i) remove the figure or ii) supply a replacement figure that complies with the CC BY 4.0 license. Please check copyright information on all replacement figures and update the figure caption with source information. If applicable, please specify in the figure caption text when a figure is similar but not identical to the original image and is therefore for illustrative purposes only.

7. We note you have included a table to which you do not refer in the text of your manuscript. Please ensure that you refer to Table 1 in your text; if accepted, production will need this reference to link the reader to the Table.

Reviewers' comments:

Reviewer's Responses to Questions

**Comments to the Author**

1. Is the manuscript technically sound, and do the data support the conclusions?

Reviewer #1: Yes

Reviewer #2: Yes

2. Has the statistical analysis been performed appropriately and rigorously? 

Reviewer #1: Yes

Reviewer #2: Yes

3. Have the authors made all data underlying the findings in their manuscript fully available?

Reviewer #1: No

Reviewer #2: Yes

4. Is the manuscript presented in an intelligible fashion and written in standard English?

Reviewer #1: Yes

Reviewer #2: No

5. Review Comments to the Author

Reviewer #1: Comments to author

Thank you for submitting your research for consideration for publication with the PLOS ONE. Overall, I found this an interesting manuscript to read.

I have a made general comments followed by some specific comments.

These have been provided in the attached document.

Reviewer #2: Comments

Overall it is an interesting work on calf morbidity and mortality in countries like Ethiopia where dairy production is unused and on its verge of growing. The study presents a good account of information on calf morbidity and mortality. However, the paper needs revision before considering it for publication. The specific comments are given below.

1. There are several grammatical flaws throughout the paper and the English language proficiency needs to be given due consideration during the revision of the paper.

2. In the introduction section:

The description of the diary production characteristics of the country and that of the study area is lacking.

In last paragraph of the introduction on lines 4-6, the author (s) mortality proportions but claimed as “rates”, which needs correction

Strong justification is not provided

3. Materials and Methods section:

The descriptions of the dairy farms used are not given

The same words are spelt different in several places. For example; “smallholder” is spelt sometimes as “sall holder” and yet the other times as “small-holder” but both are wrong; the term “cross-sectional” is also spelt differently

The author (s) claimed that two study designs were used. Cross-sectional design for collection of data by interviewing and longitudinal design for monitoring of the occurrence of morbidity and mortality. I think collection of information on those variables can be done as part of the longitudinal study. So there is no cross-sectional study

4. Results:

Some of the results do not make sense. For instance; in Table 2 the total family size presented was 160 in the second column and for the same variable in the totals column it was 984. Both do make any sense. For that matter the whole table is not important

In the same table, the chi-squared test compared the proportion of respondents in each category of the variables studied. It did not associate the variables with calf morbidity and mortality.

The table 3 cited in the text on page 10 should be table 2 not table 3

The author (s) indicated on page 12 second paragraph that 58 of the 196 calves recruited were lost- to- follow-up. That is, 29.59% of the original study population was lost. However, when greater than 10% of the study population is lost it is difficult to achieve the desired result. Especially it is difficult to make conclusions from the statistical associations observed. The author (s) should at least provide this in the limitations of the study

On page 12 last paragraph, the incidence of morbidity is given. Incidence by itself is measure morbidity so one of them is sufficient.

The cohort used was open cohort so computation of the cumulative incidence is impossible or not the actual measure of association. Similar flaws is observed on page 14

5. Discussion

Pages 21 and 22 are dedicated to comparison of morbidity and mortality proportions not rates. The objectives of the study and the findings are incidence rates. So the discussion has to be re-considered

The conclusion is too much

6. PLOS authors have the option to publish the peer review history of their article (what does this mean?). If published, this will include your full peer review and any attached files.

Reviewer #1: **Yes: **Dr. Ashley J Phipps BVSc (HonsI) GradCertVPH(EAD) MVS MVSc(Clinical) FANZCVS (Dairy Cattle Medicine and Management)

Reviewer #2: No

---

## [Author Response · Author response to Decision Letter 0]

14 Jul 2021

1. Regarding additional information

We have attached the necessary additional information that includes the questionnaire survey which is the herd-level cross-sectional study and the calf-level longitudinal study parts.

2. Regarding the minimal dataset

As we, authors, understood from the definition the minimal dataset for our manuscript includes certain parts of the values of behind hazard ratio, calf-survival days and herd-level survey responses and values we used to build graphs. Thus, we have attached these values.

3. Regarding to study area map

We have used the study area map from maps produced by the Ethiopian cartography agency for country’s national data. However, we cannot obtain permission because it is already released to use within the country. Thus, we used the second option and have removed the map from the manuscript.

4. Regarding ORCID ID

I have ORCID ID as per the PLOSE guideline for the corresponding author

5. Regarding misspelling 

 Words like small-holders and cross-sectional study have already corrected 

6. Regarding study designs

We developed two study designs for data collection. Hence, the cross-sectional the design was used to collect the general herd-level data by interviewing the respondents about past events in the snapshot, whereas the longitudinal design was used for monitoring the incidence of morbidity and mortality about six months period. Thus, we believe the aim of both study designs and collected data were different and those variables can be done differently under respective study designs.

7. Regarding the family size

In Table 1, the total family size presented was 984 which means that the total number of individuals under the family of smallholders who were interviewed for this study (N = 160).

In the Table 2, the chi-squared test compared the proportion of respondents in each category of the variables studied to the calf health and mortality problems in farms for the past 1-year event occurrence as shown in the questionnaire (N=52). Thus, we suppose these variables categories have association with calf morbidity and mortality, and they may become determinant to the outcome of interest. By the way, N = 52 is the number of calf health and mortality problems faced in the past 1year in the respective study farms.

9. Regarding lost-to-follow-up population

Of course, 29.59% of the original study population was lost. However, the study population is an open cohort and replacement of the population was made particularly for withdrawal, hence 24.5% of calves were exited in the study period due to diseases, deaths, and sales. The study focus was animal-time at-risk, so-called incidence rate, but not cumulative incidence. However, according to Martin et al. (1987) we can convert the incidence rate (True Rate) values to the cumulative incidence whether the cohort is open or closed but not neglecting actual measure of association. 

Open population: incidence risk = 1−exp(-incidence rate x length of study period). This formula is used to relate prevalence and incidence

We recorded all the outcomes and most explanatory variables from direct observations and suppose the test for association with disease outcome, however it might need cautious considerations due to the small number of death events which lower the power of the test in survival analysis.

10. Regarding the use of discussion with proportions (prevalence)

The discussion using proportions have made based on the conversion of the incidence rate (true rate) to cumulative incidence (risk rate) according to Martin et al. (1987) and Stevenson (2005) for open population computation of prevalence. With this regard the objective of this study was not only incidence rate but also cumulative incidence, because the objective as indicated in the last paragraph introduction section was designed to estimate the incidence of calf morbidity and mortality and the associated risk factors.

---

## [Decision Letter · Decision Letter 1]

25 Aug 2021

Longitudinal study of calf morbidity and mortality on smallholder farms in southern Ethiopia

PONE-D-21-01541R1

Dear Dr. Tora,

We’re pleased to inform you that your manuscript has been judged scientifically suitable for publication and will be formally accepted for publication once it meets all outstanding technical requirements.

Kind regards,

Angel Abuelo, DVM, MRes, MSc, PhD, DABVP (Dairy), DECBHM

Academic Editor

PLOS ONE

Additional Editor Comments (optional):

Reviewers' comments:

Reviewer's Responses to Questions

**Comments to the Author**

1. If the authors have adequately addressed your comments raised in a previous round of review and you feel that this manuscript is now acceptable for publication, you may indicate that here to bypass the “Comments to the Author” section, enter your conflict of interest statement in the “Confidential to Editor” section, and submit your "Accept" recommendation.

Reviewer #1: All comments have been addressed

Reviewer #2: All comments have been addressed

2. Is the manuscript technically sound, and do the data support the conclusions?

Reviewer #1: Yes

Reviewer #2: Yes

3. Has the statistical analysis been performed appropriately and rigorously? 

Reviewer #1: Yes

Reviewer #2: Yes

4. Have the authors made all data underlying the findings in their manuscript fully available?

Reviewer #1: Yes

Reviewer #2: Yes

5. Is the manuscript presented in an intelligible fashion and written in standard English?

Reviewer #1: Yes

Reviewer #2: Yes

6. Review Comments to the Author

Reviewer #1: Thanks to the authors for re-submitting your research for consideration for publication with the PLOS ONE. I would also like to thank the authors for addressing the comments previously detailed and answering the questions asked by each of the reviewers.

Reviewer #2: The authors tried to include my comments and made changes or improvements to the paper which led it to be ok for publication

7. PLOS authors have the option to publish the peer review history of their article (what does this mean?). If published, this will include your full peer review and any attached files.

Reviewer #1: No

Reviewer #2: No

---

## [Editor Report · Acceptance letter]

31 Aug 2021

PONE-D-21-01541R1 

Longitudinal study of calf morbidity and mortality on smallholder farms in southern Ethiopia 

Dear Dr. Tora:

I'm pleased to inform you that your manuscript has been deemed suitable for publication in PLOS ONE. Congratulations! Your manuscript is now with our production department. 

Kind regards, 

on behalf of

Dr. Angel Abuelo 

Academic Editor

PLOS ONE